# Vaginal Progesterone Has No Diabetogenic Potential in Twin Pregnancies: A Retrospective Case-Control Study on 1686 Pregnancies

**DOI:** 10.3390/jcm9072249

**Published:** 2020-07-15

**Authors:** Klara Rosta, Katharina Al-Bibawy, Maria Al-Bibawy, Wilhelm Temsch, Stephanie Springer, Aniko Somogyi, Johannes Ott

**Affiliations:** 1Department of Obstetrics and Gynecology, Clinical Division of Gynecologic Endocrinology and Reproductive Medicine, Medical University of Vienna, Vienna 1090, Austria; klara.rosta@meduniwien.ac.at (K.R.); n1542411@students.meduniwien.ac.at (K.A.-B.); n1542399@students.meduniwien.ac.at (M.A.-B.); stephanie.springer@meduniwien.ac.at (S.S.); 2Center for Medical Statistic and Informatic and Intelligent Systems, Medical University of Vienna, Vienna 1090, Austria; wilhelm.temsch@meduniwien.ac.at; 32nd Department of Internal Medicine, Semmelweis University, 1085 Budapest, Hungary; somogyi.aniko@med.semmelweis-univ.hu

**Keywords:** twin pregnancy, gestational diabetes mellitus, preterm birth, vaginal progesterone, high-risk pregnancy

## Abstract

Background: In this study, we aimed to investigate the incidence of gestational diabetes mellitus (GDM) in women who carried twin pregnancies and received vaginal progesterone. Methods: In this retrospective cohort study, 203 out of 1686 women with twin pregnancies received natural progesterone (200 mg/day between gestational weeks 16 + 0 and 36 + 0) vaginally for ≥ 4 weeks. The control group consisted of 1483 women with twin pregnancies without progesterone administration. Pearson’s Chi squared test, Fisher’s exact test, and Student’s *t*-test was used to compare differences between the control and the progesterone-treated groups. A multivariate binary logistic regression was performed to assess relative independent associations on the dependent outcome of GDM incidence. Results: Vaginal progesterone treatment in twin pregnancies had no significant influence on developing GDM (*p* = 0.662). Higher pre-pregnancy BMI (OR 1.1; *p* < 0.001), GDM in previous pregnancy (OR 6.0; *p* < 0.001), and smoking during pregnancy (OR 1.6; *p* = 0.014) posed an increased risk for developing GDM. Conclusion: In twin pregnancies, the use of vaginal progesterone for the prevention of recurrent preterm delivery was not associated with an increased risk of GDM.

## 1. Introduction

Gestational diabetes mellitus (GDM) is defined as glucose intolerance first recognized during pregnancy. GDM is a common disease with a reported prevalence of 2–10% in Europe (e.g., >10% in Austria) [1]. The prevalence of GDM is on the rise, which might be a consequence of delayed childbearing age as well as of the obesity epidemic secondary to sedentary lifestyle and western diet [2]. Nonetheless, the effect of iatrogenic factors such as the widespread use of progesterone administration should be also considered. Without doubt, GDM is a feared complication of many pregnancies, which can be of high clinical relevance with possible severe adverse outcomes during pregnancy, at delivery or later in life for both the mother and for the offspring. In the long run, women with GDM during pregnancy have an elevated risk of type 2 diabetes and cardiovascular disease, while children born to mothers with GDM have higher risk for developing obesity and high blood pressure already as adolescents [3].

Progestogens are widely used for the prevention of preterm birth (PTB) in both singleton and twin pregnancies [4,5,6]. Preterm birth is a leading cause of perinatal morbidity and mortality and one of the most common complications of pregnancy, with an estimated worldwide prevalence of 11.1%. In 2014, approximately 14.8 million children were born preterm worldwide. In the USA, about 12% of all births are preterm births, which, according to a 2006 estimate by the Institute of Medicine, leads to annual costs of about 26.2 billion USD (16.5 billion EUR). Preterm birth rates in 2014 ranged from 8.4% in Europe to 14.7% in North Africa [7]. According to the “Statistik Austria”, premature birth rate was 7.7% in 2018 [8]. It is important to note that although preterm survival rates have increased in high-income countries, without high quality neonatal care for preterm newborns, morbidity and mortality stay a major problem in many low or middle-income countries. For that reason, primary prevention is a major goal.

Notably, twin pregnancies account for 17% and 23% of births before 37 and 32 weeks of gestation, respectively [9]. Twin gestation elevates the risk of preterm birth by a factor of six and having a shortened cervical length <25 mm further increases this risk significantly [10]. There are several methods suggested in the literature to detect and select for twin pregnancies at high risk of preterm birth. Among these suggested predictors of spontaneous preterm birth are cervical length monitoring [11], history of previous singleton preterm delivery [12], measurement of fetal fibronectin level [13], and uterine activity monitoring [14]. Unfortunately, none of these methods brought an unequivocal success, thus, none of these methods are suggested in the follow up of multiple pregnancies. What is the problem with all those suggested screening methods? A plethora of studies exist with different cut-offs for preterm birth, varying gestational age at the screening assessment, and different cut-off values of the screening parameter measured. Moreover, those studies, which were conducted in a prospective and randomized manner, were biased by not blinding the treatment provider at the time point of applying the screening measurement. This resulted in very serious and serious risk of bias using the QUIPS checklist.

From all the above mentioned, possible predictors of preterm birth cervical length measurements seem the most promising method. The evidence in twin pregnancy suggests that cervical length is a moderate predictor of early onset spontaneous preterm birth. Clarifying which twin pregnant women are at risk of preterm birth is important, since it allows obstetricians for closer follow up and for an intervention to be offered. 

The incidence of multiple gestation is continuously increasing due mainly to the widespread use of assisted reproductive technology. Unfortunately, the large increase in twin pregnancies has resulted in an increase in spontaneous and induced PTB [15]. In the last decade, the utility of progesterone administration has been extensively studied, in particular for its effect on prolonging pregnancy and thereby ensuring better neonatal outcome. This led to a substantial increase in the use of synthetic and natural progestogens in singleton and twin pregnancies [16,17,18]. 

There are, as of yet, no effective evidence-based strategies for the prevention of preterm birth in twin pregnancies. The individual patient data meta-analysis by Romero et al. (2017) showed promising results, but other well-designed, randomized, double-blind studies are needed. In this meta-analyses, the data of six studies were included, with 303 asymptomatic twin pregnancies with a cervical length of ≤ 25 mm in the 2nd trimester. According to Romero et al., there was a significant reduction in the preterm birth rate at or under 33 gestational weeks (31.4 vs. 43.1%; RR 0.69; 95% CI 0.51–0.93, primary outcome) and an improvement in neonatal outcome, e.g., reduction in neonatal mortality (RR 0.53; 95% CI 0.35–0.81), respiratory distress syndrome (RR 0.70; 95% CI 0.56–0.89), and a reduction in births of newborns under < 1500 g (RR 0.53; 95% CI 0.35–0.80). It is interesting to note that 70.4% of the patients in this meta-analyses used 400 mg vaginal progesterone. These data suggest that women with twin pregnancies might benefit from progesterone administration in cases of shortened cervical length (<25 mm) [17]. Progesterone is a steroid hormone derived from cholesterol, which is produced by the corpus luteum during the menstrual cycle. In the period before ovulation, progesterone prepares the endometrium for a possible nidation of the fertilized egg. In the case of fertilization, the syncytiotrophoblast that is formed secretes the hormone human chorionic gonadotropin (hCG), which promotes further progesterone secretion from the corpus luteum, thereby preventing ovulation. The production of progesterone is overtaken by the placenta after about eight weeks. In early pregnancy, progesterone appears to mediate the maintenance of pregnancy. Its mode of action in the further course of pregnancy is not entirely clear. On the one hand, it inhibits the synthesis of prostaglandins, which have a contractile effect. It also inhibits the expression of genes responsible for uterine contractions, such as genes for oxytocin and prostaglandin receptors, gap junctions, and ion channels [18]. In vitro and in vivo studies have also demonstrated immunomodulatory effects on progesterone; for example, progesterone has been shown to reduce TNF alpha-mediated apoptosis of fetal membranes and to inhibit peripheral immune cells and immune responses in utero [19]. 

Depending on the method of administration, progesterone therapy can result in various adverse effects, some of them systemic. For oral administration, an increased rate of headaches, dizziness, and fatigue has been described compared with placebo. As a result of vaginal progesterone administration, the initial metabolism in the liver is bypassed, which leads to lower systemic at the same time as increased local bioavailability. This has been referred to as the “first uterine pass effect”. This also results in a reduction in the systemic adverse effects described with oral progesterone administration. Studies point to an increase in vaginal discharge when progesterone is administered vaginally [6].

Progesterone has been reported to possess diabetogenic effects [20]. While this property of progesterone has been examined in both in vitro [21,22], retrospective [23,24,25], and prospective [26,27] studies, results remain inconclusive. The majority of studies investigated the effect of 17-alpha hydroxyprogesterone caproate (17 OHPC), an intramuscular synthetic progesterone, on GDM risk, mostly in singleton pregnancies. Recently, our study group as well as Zipori et al. suggested in retrospective case control studies that vaginal progesterone administration did not affect the risk for GDM in singleton pregnancies [24,25]. Notably, the prevalence of GDM is increased in women with twin gestation [28]. It has been suggested that pregnancy itself exerts “diabetogenic” effects by increasing several, mostly placenta-derived endocrine and paracrine factors. Indeed, higher placental mass—which correlates with these hormones—might be responsible for this effect [29]. Hypothetically, exposing these women at risk to progesterone with its assumed diabetogenic effect might be of major relevance. However, there are still only very limited data about the diabetogenic effect of progesterone administration in multiple pregnancies [26] and there are no data about vaginal natural progesterone substitution in twin pregnancies. 

## 2. Methods 

### 2.1. Patients

For this retrospective study, women with twin gestations were considered eligible if they underwent prenatal care and delivered at the Department of Obstetrics and Gynecology of the Medical University of Vienna (MUW), a tertiary center for fetomaternal medicine in eastern Austria (date of delivery: from January 2004 to December 2018). Exclusion criteria included the following: pre-existent diabetes mellitus; intrauterine fetal death (IUFD) of ≥ 1 fetus before gestational week 24 + 0; multiple gestations after fetal reduction to twin pregnancies; oral or intramuscular progesterone administration; deliveries < 24 + 0 weeks; treatment with vaginal progesterone < 4 weeks; incomplete data due to unforeseen delivery in another center. Women treated with vaginal progesterone for ≥ 4 weeks from the second trimester onwards who did not fulfil exclusion criteria were included in the case group. The control group consisted of women with twin pregnancies without progesterone administration who underwent prenatal care and delivered at the Department of Obstetrics and Gynecology (MUW) in the time period matching that of the treatment group. The study was approved by the Institutional Review Board of the Medical University of Vienna (IRB number: 1745/2019).

### 2.2. Data Quality and Management

For data retrieval, ViewPoint Fetal Database 5.6.9.17^®^ software (General Electric Healthcare GmbH, Solingen, Germany) was used, which is the basic perinatology database at the department and contains prospectively collected information. We performed a number of checks to ensure accuracy and data quality: (a) data were first checked for distribution and range; (b) extreme values were double-checked using source files; (c) random checks were conducted by investigators. The main outcome measure was GDM, defined by an abnormal 75 g 2 h oral glucose tolerance test (OGTT) as evaluated according to recommendations by the International Association of Diabetes in Pregnancy Study Group (IADPSG) [30]. The OGTT was performed between gestational weeks (gw) 24 + 0 and 28 + 0 and was rated as abnormal if any single value reached or exceeded the threshold (fasting: 92 mg/dL; 1 h: 180 mg/dL; 2 h: 153 mg/dL). In addition, diagnosis of GDM later in the pregnancy was based on blood sugar monitoring values. Women with asymmetric fetal growth, polyhydramnion, and fetal macrosomia were suspected to have developed GDM and, in consequence, underwent blood sugar self-monitoring (fasting and 3 postprandial values 1 h after meals for one week). In the case that these values exceeded the normal threshold (i.e., <95 mg/dL fasting level or <140 mg/dL one hour after each meal), GDM was diagnosed [31]. 

Diagnostic criteria for GDM changed during the study period [24]. In order to capture all cases, we also included patients with missing OGTT values in the case group, who were labeled with the International Classification of Disease (ICD) diagnostic code for GDM (O24.41) in our database. We collected the following information from both groups: maternal age; pre-pregnancy body mass index (BMI); mode of conception; family history of diabetes mellitus; gravidity and parity; GDM in a previous pregnancy; gestational age; mode of delivery and birthweight of the children; the use of betamethasone (12 mg single course) for lung maturation and tocolysis with atosiban in twin pregnancies at risk of preterm delivery < 34 + 0 gw in the evaluated pregnancy (the latter medications are suggested to have an the influence on the development of GDM) [29]. 

### 2.3. Routine Care

All women underwent a first trimester ultrasound scan as well as an anomaly ultrasound scan at 20 + 0 to 24 + 0 gw. From the 16th gestational week on, twin pregnancies were monitored every 2–4 weeks for cervical length and fetal biometrical data. In addition, we performed the following assessments: fetal biometry, evaluation of the placenta, amniotic fluid, and urine strip test including glucose. These tests were carried out to ensure that we had the same chance of detecting late GDM development, i.e., occurrence after the routine OGTT in both groups [32,33]. Women with cervical length ≤ 25 mm at or after 16th gw were administered 200 mg vaginal progesterone until 36 + 0 gw. 

In routine care, women with GDM were consulted by both obstetricians and diabetologists. In case of a pathologic OGTT, first line intervention consisted of intensified lifestyle modification, including medical nutrition therapy. All patients were trained in capillary blood glucose monitoring and were informed over glycemic treatment targets. Follow up visits were scheduled every other week and blood glucose levels were reviewed during each appointment. Upon non-achievement of blood glucose targets (i.e., <95 mg/dL fasting level or <140 mg/dL one hour after each meal), insulin treatment was initiated at any point in time [31,34].

### 2.4. Statistical Analysis

Nominal variables are reported as numbers and frequencies, and continuous variables as means and standard deviations (SD). Statistical analysis was performed using Pearson’s Chi squared test, Fisher’s exact test, and Student’s *t*-test as appropriate on the basis of data distributions to compare differences between the control and the progesterone-treated groups. A multivariate binary logistic regression was tested using the following independent variables which are considered to influence the development of GDM: pre-pregnancy BMI, family history of diabetes mellitus, tocolysis, and lung maturation with betamethasone to assess relative independent associations on the dependent outcome of GDM incidence. For this analysis, odds ratios (OR) with the according 95% confidence intervals (95% CI), *p*-values of the likelihood ratio test, and Nagelkerke’s R^2^ for the goodness of fit were calculated. Statistical analysis was performed using the open-source statistical package, IBM SPSS 23. Differences were considered statistically significant if *p* < 0.05.

## 3. Results

### 3.1. General Characteristics

A total of 1974 women with twin pregnancies underwent prenatal care and delivered at the department during the study period. From these, *n* = 8 women were excluded due to pre-existent diabetes mellitus; *n* = 55 due to IUFD of ≥ 1 fetus before SSW 24 + 0; *n* = 15 due to multiple gestations after fetal reduction to twin pregnancies; *n* = 141 due to oral or intramuscular progesterone administration; *n* = 35 due to delivery < 24 + 0 weeks; *n* = 25 due to treatment for < 4 weeks with vaginal progesterone; *n* = 9 due to incomplete major data due to emergency delivery in another center. Accordingly, 1686 patients remained. Of these, 203 women had been treated with vaginal progesterone 200 mg/day for ≥ 4 weeks from the second trimester on, whereas 1483 women did not receive progesterone treatment.

### 3.2. Progesterone and Gestational Diabetes Mellitus

Women were aged 32.6 ± 6.0 years at delivery with a pre-pregnancy BMI of 23.9 ± 5.0 kg/m^2^. In total, 311 patients (18.4%) had developed GDM. Table 1 shows the basic patient and pregnancy characteristics of the women without (*n =* 1483) and with (*n =* 203) progesterone treatment. In women treated with vaginal progesterone, the therapy was initiated at a mean gestational age of 18.2 ± 8.8 weeks (minimum 16 + 0, maximum 32 + 0 weeks) and mean treatment duration consisted of 98 days (range: 35–203 days). Notably, in 134 women (66.0%), the initiation of progesterone administration preceded the OGTT. Maternal age was higher in the progesterone group. Moreover, administration of betamethasone as well as tocolysis with atosiban was performed more often in patients with vaginal progesterone treatment (63.5% vs. 38.4%, and 64.0% vs. 37.2%, respectively, both *p ≤* 0.001). Other patient- and pregnancy parameters did not differ between the two groups. Notably, we found no significant difference in the incidence of GDM in women carrying twin pregnancies with vs. without vaginal progesterone treatment (19.7% vs. 18.3% respectively; *p =* 0.662).

### 3.3. Factors Influencing the Development of Gestational Diabetes Mellitus

Since several factors are assumed to influence the development of GDM, a multivariate binary regression model was applied (Table 2). In this analysis, higher pre-pregnancy BMI (OR 1.091, *p <* 0.001) as well as previous pregnancy with GDM (OR 6.025, *p <* 0.001) were associated with higher risk for GDM. Notably, vaginal progesterone treatment was equally frequent in women with and without GDM (12.9% vs. 11.9%, *p =* 0.313). For this model, a Nagelkerke R^2^ of 0.117 was found.

Patients who had conceived via IVF might have received high-dose luteal support with progestogens within the first trimester. Hypothetically, this could have influenced the results. Thus, a sub-analysis of women who had conceived naturally was performed. With the use of this multivariate binary regression model, the same results were found (Appendix A). Nagelkerke’s R^2^ was 0.146.

## 4. Discussion

In this large, retrospective, case-control study of women with twin pregnancies, we could not demonstrate an association between the use of vaginal progesterone in the second trimester and increased risk of GDM development. The clinical relevance of our findings is highlighted by the widespread use of vaginal progesterone for the prevention of preterm birth in twin pregnancies, which is mainly based on the results of a meta-analysis of six randomized trials of women with twin gestations and mid-trimester cervical length ≤ 25 mm. This study revealed that vaginal progesterone significantly reduced preterm birth (defined as < 33 weeks) compared with no treatment/placebo (RR 0.69, 31% vs. 43%). The relative risk of neonatal death, respiratory distress syndrome, and birth weight < 1500 g was also reduced significantly, on average by 30–50% [17].

Only few studies analyzed the effect of progesterone substitution on carbohydrate metabolism and on the incidence of GDM. Some studies suggested that intramuscular administration of 17 OHPC might pose a risk of increasing the incidence of GDM [20,23]; on the other hand, a secondary analysis of two randomized placebo controlled trials of 17 OHPC found no association with higher rates of GDM [26]. A recent meta-analysis by Eke et al. examined women with singleton pregnancies and a history of preterm birth, who received either 17 OHPC or placebo, for the occurrence of GDM. The intervention group showed a significantly increased risk of GDM (10.9% vs. 6.1%). The meta-analysis included a total of six studies—two randomized controlled and four cohort studies. When only the two randomized controlled trials were examined, a significant result could not be maintained [35].

To the best of our knowledge, so far, only a single study investigated the diabetogenic effect of progesterone treatment in women with twin pregnancies. This secondary analysis of two double-blind randomized placebo-controlled trials of 17 OHPC, an intramuscularly administered form of progesterone, given to women at risk for preterm delivery, showed similar GDM rates in treated and untreated women (7.4% vs. 7.6%; *p* = 0.94, respectively) [26]. This average GDM rate is considerably lower compared to the rate of 18.44% found in our dataset. The high incidence of GDM in our cohort (progesterone treated vs. control 40/203, 19.7% vs. 271/1483 18.3% respectively; *p* = 0.622) might be explained by tertiary patient care provided at our center which might have led to accumulation of GDM patients which are high-risk pregnancies. The incidence of GDM in our study is comparable to the overall rate (17.8%; range 9.3–25.5%) reported by the 15 participating centers in the HAPO Study using the IADPSG criteria [36]. However, the incidence of GDM may be even greater in twin gestations, a known independent risk factor for GDM [37]. Of note, a wide range (7–20%) of GDM has been reported for twin pregnancies in the literature [28,38,39,40] and the risk of developing GDM was always increased compared to singleton pregnancies. 

Multiple gestation itself can elevate the prevalence of GDM through multiple mechanisms. One can postulate that the rapidly rising estrogen and progesterone levels alter glucose homeostasis in pregnancy. Progesterone is known to impair insulin binding to its receptor and to impair glucose transport. In vitro progesterone inhibits the proliferation and facilitates apoptotic changes in pancreatic islet cells [21]. The natural course of insulin sensitivity and resistance changes in the course of pregnancy. While in the first trimester, the level of fasting serum insulin is similar to that in non-pregnant women, fasting serum insulin levels increase significantly during the second and third trimesters, due to the rise of diabetogenic hormones such as human placental lactogen, human chorionic gonadotrophin (HCG), growth hormone, and cortisol, which results in reduced insulin sensitivity, hyperinsulinemia [41]. This effect might be even more relevant in twin gestations, where placental mass is usually higher, thus, it is more capable of secreting these hormones, not to mention the possible effect of artificial reproductive methods on carbohydrate metabolism. Moreover, in our study, a high number of women smoked during pregnancy. Prenatal smoking has been reported to be associated with higher odds of GDM [42]. We consider neither the mean age of 32.6 years nor the mean pre-pregnancy BMI of about 23.7 kg/m^2^ to have made a substantial contribution to the GDM risk.

Data on the effect of vaginally administered natural progesterone and the occurrence of GDM in twin pregnancies are scarce. One can only refer to studies in singleton pregnancies. Notably, no diabetogenic effect of vaginal progesterone has been found in small retrospective studies [24,25]. In addition, a recent meta-analysis published in 2019 reached the same conclusion that vaginal progesterone administration would not increase the risk of developing GDM in singleton pregnancies [43], which is in line with our results in twin pregnancies. Despite the postulated diabetogenic effect of progesterone, the vaginal route of administration might play a critical role, since hypothetically, the mainly local effect of vaginal progesterone on cervical tissue might denote a smaller effect on carbohydrate metabolism. Although we did not measure serum levels of progesterone in our study, increases in progesterone serum levels following topical application have not been reported [44].

Our study is limited by its retrospective design and the long time period it embraces. Since the introduction of the HAPO screening method, the incidence of GDM has increased [36]. This is due to the lower plasma glucose values in the diagnostic criteria (fasting plasma glucose exceeds 95 mg/dL, 1 h 180 mg/dL, and 2 h 155 mg/dL after 75 gr glucose loading vs. 92 mg/dL, 1 h 180 mg/dL, and 2 h 153 mg/dL by OGTT) [30,34,45]. In order to correct for this methodological problem, we classified patients for GDM not only based on their ICD diagnosis in the database, but also based on their OGTT values. This was necessary to unify the diagnosis criteria for GDM between 2004 and 2018. Altogether, thirty-one patients were reclassified as GDM according to the HAPO criteria. However, the original OGTT data were missing in 652 cases (38.7%) and, thus, the ICD diagnosis in the database had to be used, which is a study limitation. In addition, the results of our study are limited by an observation bias due the fact that the group of women who received vaginal progesterone underwent regular control check-ups more often, i.e., every other week as compared to those not receiving vaginal progesterone. Increased frequency of follow up visits may have, thus, influenced the detection of GDM after regular OGTT screening based on signs such as glycosuria, polyhydramnion, disproportional growth or fetal abdominal circumference [34]. Notably, one-third of women started with progesterone treatment after the OGTT. Nevertheless, this bias is likely negligible as the Austrian outpatient care system necessitates a detailed ultrasound examination between gestational weeks 30 and 34, to identify signs of pathology with consequent referral of affected patients to the delivery center. The results of our study underline the effectivity of this well-functioning outpatient care system, as the incidence of GDM in the control group was not found to be lower compared to that reported in international studies [36]. Moreover, the objectives of this study did not include assessing the efficacy of vaginal progesterone treatment in the prevention of preterm birth and the design did not incorporate a detailed evaluation of fetal outcomes. Maternal race or ethnicity are not routinely recorded in our database, which precluded stratification of data based on these factors. Last not least, our dataset lacks information on patients’ educational level, which we consider a minor limitation.

In conclusion, our results imply that vaginal progesterone treatment for cervical shortening in twin pregnancies does not seem to increase the rate of GDM development. Further prospective studies are needed to confirm the results of this retrospective cohort study and to evaluate the diabetogenic effect of widely used natural vaginal progesterone on populations at high risk for GDM.

## Figures and Tables

**Table 1 jcm-09-02249-t001:** Patient and pregnancy characteristics of women with and without vaginal progesterone treatment.

	Women with Vaginal Progesterone Treatment (*n* = 203)	Women without Vaginal Progesterone Treatment (*n* = 1483)	*p*-Value
Maternal age (years) *	33.2 ± 6.3	32.1 ± 6.0	0.015
Maternal age > 37 years ^#^	53 (26.1%)	240 (16.2%)	<0.001
Pre-pregnancy BMI kg/m2 *	23.5 ± 4.6	24.0 ± 5.0	0.340
Previous GDM ^#^	3 (1.5%)	46 (3.1%)	0.196
Use of atosiban for tocolysis ^#^	130 (64.0%)	552 (37.2%)	<0.001
Use of betamethason for lung maturation ^#^	129 (63.5%)	570 (38.4%)	<0.001
Gestational age at delivery (completed weeks) *	35.31 ± 2.98	35.47 ± 3.37	0.545
Preterm delivery < 37 + 0 ^#^	105 (51.7%)	688 (46.4%)	0.341
Preterm delivery < 35 + 0 ^#^	56 (27.6%)	401 (27.0%)	0.922
Preterm delivery < 32 + 0 ^#^	29 (14.3%)	182 (12.3%)	0.673
Delivery by caesarean section ^#^	186 (91.6%)	1335 (90.0%)	0.733
Birthweight (g) *	2076.9 ± 594.53	2069.31 ± 655.12	0.906
Incidence of GDM ^#^	40 (19.7%)	271 (18.3%)	0.622
Gravity	1.99 ± 1.22	2.13 ± 1.44	0.175
Parity	1.45 ± 0.84	1.71 ± 1.07	0.001
Smoking	17 (8.4%)	155 (10.5%)	0.232
Beginning of therapy	18.16 ± 8.76		
Duration of therapy	15.19 ± 7.65		

Data are presented as * mean ± standard deviation or ^#^ numbers (percentage); BMI—body mass index; GDM—gestational diabetes mellitus.

**Table 2 jcm-09-02249-t002:** Results of the binary logistic regression model for the prediction of gestational diabetes mellitus development.

	Women with GDM (*n* = 311)	Women without GDM (*n* = 1375)	OR (95% CI)	*p*
Vaginal progesterone treatment	40 (12.9)	163 (11.9)	1.228 (0.824;1.830)	0.313
Maternal age (years)	32.7 ± 5.3	32.1 ± 6.2	1.025 (1.000;1.050)	0.047
Pre-pregnancy BMI (kg/m^2^)	26.2 ± 6.1	23.4 ± 4.5	1.090 (1.063;1.117)	<0.001
Family history of diabetes	105 (33.8)	321 (23.3)	1.282 (0.961;1.712)	0.092
Previous pregnancy with GDM	31 (10.0)	18 (1.3)	6.025 (3.153;11.514)	<0.001
Betamethasone 12 mg 2x within 24 h	112 (36.0)	587 (42.7)	1.488 (0.790;2.804)	0.219
Atosiban * for at least 48 h	105 (33.8)	577 (42.0)	0.563 (0.295;1.073)	0.081
Current smoking	48 (15.4)	124 (9.0)	1.639(1.107; 2.428)	0.014

BMI—body mass index; OR—odds ratio, 95%CI: 95% confidence interval. Multivariate analyses included all parameters significant in the univariate analysis; * In the atosiban group, women received a bolus injection of 6.75 mg intravenous in 1 min, followed by 18 mg/h for 3 h, followed by a maintenance dosage of 6 mg/h for 45 h.

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
