# Peer review of "Vaginal Progesterone Has No Diabetogenic Potential in Twin Pregnancies: A Retrospective Case-Control Study on 1686 Pregnancies"

_jcm, 2020, doi:10.3390/jcm9072249_

Round 1

Reviewer 1 Report

The study aimed to evaluate whether vaginal progesterone in twin pregnancy is associated with the development of gestational diabetes mellitus. As a retrospective cohort study, the authors were able to answer the question. A few minor concerns need to be addressed:

  1. In conclusion, the authors state "does not lead to". It is inappropriate to state causation in a retrospective study. It is a hypothesis generator, will need to be evaluated in an RCT. 
  2. Need more demographic data in the Table 1 including gradivity, parity, smoking, education and race.
  3. Need more data presented for the timing of progesterone supplementation and testing for GDM
  4. Was there any missing data? How was it handled?

Author Response

Dear Editor,

Dear Reviewers,

my co-authors and I thank you for your valuable comments that helped us to improve our manuscript. We carefully revised our manuscript according to your suggestions and concerns. Please find a point-by-point answer to your comments below. In addition to these revisions, we added several paragraphs on general issues about GDM and twins according to the editor`s recommendation. The main text of the manuscript now nearly reaches 4.000 words. However, since it is not a review, we did not include much more information outside the manuscript’s main topic. If this is still considered insufficient, we shall be happy to add more content in the course of a future revision process.

We hope that the revisions will make our manuscript acceptable for publication in “Journal of Clinical Medicine”.

Respectfully yours,

Johannes Ott

Reviewer #1

The study aimed to evaluate whether vaginal progesterone in twin pregnancy is associated with the development of gestational diabetes mellitus. As a retrospective cohort study, the authors were able to answer the question. A few minor concerns need to be addressed:

Comment 1: In conclusion, the authors state "does not lead to". It is inappropriate to state causation in a retrospective study. It is a hypothesis generator, will need to be evaluated in an RCT. 

Answer 1: Thank you for this correction. The sentence was rewritten accordingly: lines 239-241

“In conclusion, our data suggest that vaginal progesterone treatment for cervical shortening in twin pregnancies does not seem to increase the rate of GDM development. Further prospective studies are needed to confirm the result of this retrospective cohort study and to to evaluate the diabetogenic effect of widely used natural vaginal progesterone on populations at high risk for GDM”

Comment 2: Need more demographic data in the Table 1 including gradivity, parity, smoking, education and race.

Answer 2:

Gravidity, parity and smoking during pregnancy was assessed in our study and now included in Table 1.  Unfortunately, educational level and race is not documented reliably in our system. Therefore, these parameters cannot be assessed/ included in Table 1. See correction in Table 1.

Moreover, we added the following sentence to the Discussion Section: “Last not least, our data set lacks information on patients’ educational level and race which we consider a minor limitation.”

Comment 3: Need more data presented for the timing of progesterone supplementation and testing for GDM

Answer 3:

We thank the reviewer for this important comment. We added the following sentences to the manuscript:

Results (Line 253-256): “In women treated with vaginal progesterone, the treatment was initiated at a mean gestational age of 18.2 ± 8.8 weeks (minimum 16+0, maximum 32+0 weeks) and mean treatment duration consisted of 98 days (range: 35-203 days). Notably, in 134 women (66.0%), the initiation of progesterone administration preceded the OGTT.”

Discussion Line 375: “[Increased frequency of follow-up visits may have thus influenced the detection of GDM after regular OGTT screening based on signs such as glycosuria, polyhydramnion, disproportional growth or fetal abdominal circumference [21].] Notably, one third of women started with progesterone treatment after the OGTT. Nevertheless, […]”

Comment 4: Was there any missing data? How was it handled?

Answer 4: We thank the reviewer for this comment:

Patients who took part in the ambulatory care for twin pregnancies and delivered at our department were included in this study. Only a scarce number (n= 9) of pregnant women delivered outside of our clinic. These patients were excluded from the analysis due to missing obstetric data. Fetal weight, mode of delivery, age of the mother, gestational week at delivery, gravidity, parity, and use of tocolysis and lung maturation was reliably documented in our system. These data were not missing.

Moreover, please find the following new information:

Discussion (paragraph on study limitations Line 369): “However, the original OGTT data were missing in 652 cases (38.7%) and, thus, the ICD diagnosis in the database had to be used which is a study limitation.”

Reviewer 2 Report

A retrospective study examining the frequency of gestational diabetes in twin gestations treated with vaginal progesterone for the prevention of preterm birth.

What was the relationship regarding the timing of the glucose tolerance testing and the administration of the vaginal progesterone?

Line 74- what does SSW mean?

Lines 109-112 - The authors suggest the diagnosis of gestational diabetes would have been made after the glucose tolerance test

"These follow-up examinations included fetal biometry,  evaluation of the placenta, amniotic fluid, and a dip stick urine test including glucose. Thereby, cases of late GDM development, i.e. after the routine OGTT, should have become equally evident in both groups."

How was the diagnosis of gestational diabetes made based on the evidence above?

Author Response

Dear Editor,

Dear Reviewers,

my co-authors and I thank you for your valuable comments that helped us to improve our manuscript. We carefully revised our manuscript according to your suggestions and concerns. Please find a point-by-point answer to your comments below. In addition to these revisions, we added several paragraphs on general issues about GDM and twins according to the editor`s recommendation. The main text of the manuscript now nearly reaches 4.000 words. However, since it is not a review, we did not include much more information outside the manuscript’s main topic. If this is still considered insufficient, we shall be happy to add more content in the course of a future revision process.

We hope that the revisions will make our manuscript acceptable for publication in “Journal of Clinical Medicine”.

Respectfully yours,

Johannes Ott

Reviewer #2

A retrospective study examining the frequency of gestational diabetes in twin gestations treated with vaginal progesterone for the prevention of preterm birth.

Question 1: What was the relationship regarding the timing of the glucose tolerance testing and the administration of the vaginal progesterone?

Answer 1: We thank the reviewer for this important comment. We added the following sentences to the manuscript:

Results (Line 253-256): “In women treated with vaginal progesterone, the treatment was initiated at a mean gestational age of 18.2 ± 8.8 weeks (minimum 16+0, maximum 32+0 weeks) and mean treatment duration consisted of 98 days (range: 35-203 days). Notably, in 134 women (66.0%), the initiation of progesterone administration preceded the OGTT.”

Discussion Line 375: “[Increased frequency of follow-up visits may have thus influenced the detection of GDM after regular OGTT screening based on signs such as glycosuria, polyhydramnion, disproportional growth or fetal abdominal circumference [21].] Notably, one third of women started with progesterone treatment after the OGTT. Nevertheless, […]”

Question 2: Line 74- what does SSW mean?

Answer 2: We are very sorry for this mistake. “SSW” means “gestational week” in German. We corrected this mistake in the text.

Comment 3: Lines 109-112 - The authors suggest the diagnosis of gestational diabetes would have been made after the glucose tolerance test.

"These follow-up examinations included fetal biometry, evaluation of the placenta, amniotic fluid, and a dip stick urine test including glucose. Thereby, cases of late GDM development, i.e. after the routine OGTT, should have become equally evident in both groups."

How was the diagnosis of gestational diabetes made based on the evidence above?

Answer 3: In the revised manuscript, we describe the diagnosis of GDM after OGTT more in detail in the Methods Section: “In addition, diagnosis of GDM later in the pregnancy was based on blood sugar monitoring values. Women with asymmetric fetal growth, polyhydramnion, fetal macrosomia were suspected to have developed GDM and, in consequence, underwent blood sugar self-monitoring (fasting and 3 postprandial values 1 hour after meals for one week). In case that these values exceeded the normal threshold (i.e. <95 mg/dl fasting level or <140 mg/dl one hour after each meal), GDM was diagnosed [new ref].”

We also added the following reference: “Blumer I, Hadar E, Hadden DR, Jovanovič L, Mestman JH, Murad MH, u. a. Diabetes and Pregnancy: An Endocrine Society Clinical Practice Guideline. J Clin Endocrinol Metab. November 2013;98(11):4227–49.”

Round 2

Reviewer 1 Report

The manuscript has improved. Congratulations to authors for the hardwork.